# RHΔ*gra17*Δ*npt1* Strain of *Toxoplasma gondii* Elicits Protective Immunity Against Acute, Chronic and Congenital Toxoplasmosis in Mice

**DOI:** 10.3390/microorganisms8030352

**Published:** 2020-03-01

**Authors:** Qin-Li Liang, Li-Xiu Sun, Hany M. Elsheikha, Xue-Zhen Cao, Lan-Bi Nie, Ting-Ting Li, Tao-Shan Li, Xing-Quan Zhu, Jin-Lei Wang

**Affiliations:** 1State Key Laboratory of Veterinary Etiological Biology, Key Laboratory of Veterinary Parasitology of Gansu Province, Lanzhou Veterinary Research Institute, Chinese Academy of Agricultural Sciences, Lanzhou 730046, China; liangqinli701@163.com (Q.-L.L.); sunlixiu1026@163.com (L.-X.S.); 18702034757@189.com (X.-Z.C.); nielanbi188@163.com (L.-B.N.); litt866@163.com (T.-T.L.); m18394490562@163.com (T.-S.L.); xingquanzhu1@hotmail.com (X.-Q.Z.); 2Faculty of Medicine and Health Sciences, School of Veterinary Medicine and Science, University of Nottingham, Sutton Bonington Campus, Loughborough LE12 5RD, UK; hany.elsheikha@nottingham.ac.uk; 3Jiangsu Co-innovation Center for the Prevention and Control of Important Animal Infectious Diseases and Zoonoses, Yangzhou University College of Veterinary Medicine, Yangzhou 225009, China

**Keywords:** *Toxoplasma gondii*, immunization, dense granule protein 17, novel putative transporter 1, live-attenuated vaccine

## Abstract

In the present study, a dense granule protein 17 (*gra17)* and novel putative transporter (*npt1*) double deletion mutant of *Toxoplasma gondii* RH strain was engineered. The protective efficacy of vaccination using RHΔ*gra17*Δ*npt1* tachyzoites against acute, chronic, and congenital toxoplasmosis was studied in a mouse model. Immunization using RHΔ*gra17*Δ*npt1* induced a strong humoral and cellular response, as indicated by the increased levels of anti-*T. gondii* specific IgG, interleukin 2 (IL-2), IL-10, IL-12, and interferon-gamma (IFN-γ). Vaccinated mice were protected against a lethal challenge dose (10^3^ tachyzoites) of wild-type homologous (RH) strain and heterologous (PYS and TgC7) strains, as well as against 100 tissue cysts or oocysts of Pru strain. Vaccination also conferred protection against chronic infection with 10 tissue cysts or oocysts of Pru strain, where the numbers of brain cysts in the vaccinated mice were significantly reduced compared to those detected in the control (unvaccinated + infected) mice. In addition, vaccination protected against congenital infection with 10 *T. gondii* Pru oocysts (administered orally on day 5 of gestation) as shown by the increased litter size, survival rate and the bodyweight of pups born to vaccinated dams compared to those born to unvaccinated + infected dams. The brain cyst burden of vaccinated dams was significantly lower than that of unvaccinated dams infected with oocysts. Our data show that *T. gondii* RHΔ*gra17*Δ*npt1* mutant strain can protect mice against acute, chronic, and congenital toxoplasmosis by balancing inflammatory response with immunogenicity.

## 1. Introduction

*Toxoplasma gondii*, an intracellular protozoan parasite, is the etiological agent of toxoplasmosis that can affect nearly all warm-blooded animals and humans [1,2]. About one-third of the world’s population has been estimated to be chronically infected with *T. gondii,* and infection is generally asymptomatic in immunocompetent individuals [3,4]. However, *T. gondii* infection in immunocompromised individuals, such as those with HIV/AIDS, organ transplants, or malignancies, can cause severe illness or even death [5]. Furthermore, primary infection with *T. gondii* during pregnancy can cause neonatal blindness, mental retardation, retinitis, and cognitive impairment [5,6]. Current anti-parasitic medications used to control toxoplasmosis have side effects and lack the ability to eliminate *T. gondii* tissue cysts [7]. This makes the development of new and efficacious vaccines against various clinical forms of toxoplasmosis a major goal for the improvement of global health.

Vaccines are the most effective means of disease prevention and can minimize the reliance on and side effects of chemotherapeutics. Various strategies have been used to develop a toxoplasmosis vaccine, such as DNA vaccines, protein vaccines, inactivated vaccines, epitope vaccines, and live-attenuated vaccines [8]. The only commercially available live-attenuated vaccine (Toxovax^®^), derived from S48 *T. gondii* tachyzoites, is used to protect against abortion in sheep [9,10,11]. However, this vaccine has limitations, such as the potential for reversion to the pathogenic phenotype [12]. Recent attempts to develop a toxoplasmosis vaccine have focused on developing live-attenuated *T. gondii* strains via deletion of key genes, such as RHΔ*gra17* and PruΔ*cdpk2* strains [13,14].

Dense granule proteins (GRAs), such as the *gra17* of *T. gondii*, play roles in the biogenesis and maturation of the parasitophorous vacuole (PV) and parasitophorous vacuole membrane (PVM), and nutrient acquisition [15,16]. Live attenuated RHΔ*gra17* mutant strain can induce a protective immune response against *T. gondii* infection and reduce the parasite cyst’s burden in the brain of vaccinated mice and their pups [13]. In recent years, several ‘novel putative transporters’ (NPTs) were identified, such as the *NPT* protein of *T. gondii* (Tg*npt1*), a selective transporter of arginine, which plays a crucial role in the growth and virulence of *T. gondii* [17].

Given the important role of *gra17* and *npt1* in the virulence and growth of *T. gondii*, we hypothesized that genetic attenuation of *T. gondii* by inducing double deletion of *gra17* and *npt1* genes may produce a safer and immunogenic attenuated vaccine candidate. In this paper, we describe the construction and characterization of the efficacy *T. gondii* RHΔ*gra17*Δ*npt1* double mutant strain in the prevention of acute, chronic, and congenital forms of toxoplasmosis. 

## 2. Materials and Methods

### 2.1. Mice

Seven- to nine-week-old male and female Kunming mice were purchased from the Center of Laboratory Animals of Lanzhou Veterinary Research Institute (LVRI). This study was approved by the Animal Ethics Committee of LVRI, Chinese Academy of Agricultural Sciences. Kunming mice were used because of their ability to produce more pups per litter [18], providing better means to examine the protection induced by vaccination against congenital infection. In addition, compared to BALB/c and C57BL/6 mice, Kunming mice had more susceptibility to acute and chronic *T. gondii* infection [11].

### 2.2. Parasites

Tachyzoites of *T. gondii* type 1 RH strain, and type ToxoDB#9 (PYS and TgC7) strains were maintained in vitro in monolayers of human foreskin fibroblasts (HFFs, ATCC SCRC-1041). HFFs were cultured in Dulbecco’s modified Eagle’s medium (DMEM), containing 2% fetal bovine serum (FBS), as previously described [19]. Freshly egressed tachyzoites were harvested from heavily infected HFFs using 3-µm polycarbonate membranes. Cysts of *T. gondii* type 2 Pru strain were maintained in Kunming mice and obtained from brain homogenates as previously described [13,14]. Oocysts of type 2 Pru strain were prepared as previously described [20]. Briefly, a 10-week-old, specific-pathogen-free kitten was infected orally with ~100 cysts, which were obtained from brain homogenates of Kunming mice that had been infected with the Pru strain for one month. The cat feces were examined daily for the presence of oocysts, and once detected, oocysts were isolated as previously described [21]. To induce sporulation, the purified oocysts were suspended in 2% sulfuric acid and incubated on a shaker at ambient temperature for one week. Once sporulation has been completed, the oocysts were washed twice with 0.85% saline and stored in 2% sulfuric acid at 4˚C until use.

### 2.3. Engineering of RHΔgra17Δnpt1 Mutant Strains

Mutant strains were constructed using the clustered regularly interspaced short palindromic repeats (CRISPR)/Cas9 approach. All guide RNAs and primers used in this study are listed in Appendix A. The *npt1* targeting CRISPR plasmid was constructed by replacing the UPRT targeting guide RNA (gRNA) in pSAG1::CAS9-U6::sgUPRT with an *npt1* targeting gRNA by site-directed mutagenesis, as described previously [13]. The CRISPR plasmid and the *Ble* amplicons that were amplified from a pSAG1-Ble plasmid were co-transfected into RHΔ*gra17* strain. Subsequently, transfectants were selected with phleomycin (for Ble), and single-cloned by limiting the dilution in 96-well tissue culture plates seeded with HFF monolayers. Single positive clones were identified by diagnostic PCRs. The positive parasite mutant strain was named RHΔ*gra17*Δ*npt1*.

### 2.4. Optimization of the Vaccination Dose

To evaluate the in vivo virulence of RHΔ*gra17*Δ*npt1*, serial doses (10^2^, 10^3^, 10^4^, 10^5^, 10^6^ tachyzoites) of RHΔ*gra17*Δ*npt1* strain or 10^2^ tachyzoites of wildtype (WT) RH strain were intraperitoneally (i.p.) injected into mice. The infected mice were monitored daily for the signs of toxoplasmosis, and the mortality was recorded for 30 days. We also examined the ability of RHΔ*gra17,* RHΔ*npt1,* and RHΔ*gra17*Δ*npt1* to proliferate in mice. To achieve this, each mouse was infected i.p. with 10^3^ tachyzoites (five mice/strain), and five days later, the parasite burden in the peritoneal fluid was examined by plaque assay as described previously [22]. Briefly, mice were humanely sacrificed, and peritoneal cells were harvested by lavage with phosphate buffered saline (PBS) containing 1% FBS. The harvested peritoneal cells were mechanically disrupted using a 17-gauge needle syringe in order to liberate intracellular parasites. Then 100 µL freshly liberated parasites were used to infect confluent HFF monolayers maintained in RPMI1640 medium for seven days. Then, *T. gondii*-infected HFFs were fixed and stained, and the sizes and numbers of plaques formed by the growing tachyzoites were determined using the plaque assay as described previously [22].

### 2.5. Immune Responses Induced by Immunization with RHΔgra17Δnpt1

Mice were vaccinated i.p. with 10^6^ tachyzoites of RHΔ*gra17*Δ*npt1* or mock-vaccinated in a total volume of 200 µl PBS (Appendix A). Serum samples collected at two months after immunization were used to detect anti-*T. gondii* total IgG and subclasses of IgG (IgG1, and IgG2a) antibodies by enzyme-linked immunosorbent assay (ELISA), as described previously [13,14]. The levels of cytokines in mouse splenocyte culture supernatants were determined as previously described [13,14]. Briefly, two months post-immunization, mice were sacrificed and splenocyte cultures were prepared. Splenocyte cultures were stimulated in vitro with 10 µg/mL *T. gondii* soluble tachyzoite antigen (STAg), and splenocyte culture supernatants were collected to measure the levels of secreted cytokines using ELISA. The interleukin 2 (IL-2) was measured at 24 h post-incubation, IL-10 at 72 h post-incubation, and IL-12 and interferon-gamma (IFN-γ) at 96 h post-incubation, as per the manufacturer’s recommendations (eBioscience^®^ Bender MedSystems GmbH, Austria).

### 2.6. Protection of Mice Against Acute and Chronic Infection

Two months after immunization, mice were injected i.p. with 10^3^ type 1 RH, ToxoDB#9 TgC7 or PYS strains in order to evaluate the protection of mice against acute *T. gondii*. ToxoDB#9 genotype, the predominant genotype in China, has a similar virulence to type 1 RH [23]. To evaluate the protective efficacy of RHΔ*gra17*Δ*npt1* against oral cyst and oocyst challenges, 10 or 100 of tissue cysts or oocysts of type 2 Pru strain were used to infect mice. The infected mice were observed daily and the development of clinical signs of toxoplasmosis, and survival of infected mice were recorded for 30 days. We also investigated the ability of RHΔ*gra17*Δ*npt1* to protect against chronic infection. The vaccinated mice and control mice were euthanized at 30 days after infection, and their brains were individually homogenized in 1 mL of PBS, and the parasite cyst’s burden in the brain was determined as described previously [13,14].

### 2.7. Protection Against Congenital Transmission

Two months after immunization with 10^6^ RHΔ*gra17*Δ*npt1* tachyzoites, two female mice were housed in a cage with a male mouse and examined every 12 h for the presence of vaginal plugs. The date of the presence of the vaginal plug was considered as day 1 of gestation. Mice were orally inoculated with 10 oocysts of Pru strain on day 5 of gestation. The control groups consisted of six unvaccinated + uninfected mice (negative control) and six unvaccinated mice orally infected with 10 oocysts (positive control). The protective efficacy of RHΔ*gra17*Δ*npt1* strain against congenital *T. gondii* infection was determined by analyzing the litter size and survival rate of the naturally delivered pups at birth and at 30 days old. The bodyweight of pups at 30 days old was also examined. Furthermore, the parasite cyst’s burden in the infected dams and their pups was assessed in the pups at 30 days of age, and in their dams at 30 days after delivery.

### 2.8. Statistical Analysis

Statistical analysis for the level of the differences of antibodies, cytokines, and parasite cyst burdens between mouse groups were performed using two-tailed, unpaired Student t-test (for comparing means between two groups), and one-way ANOVA analysis (for comparing means between ≥ three groups). The Mantel-Cox log-rank test was used for comparing the survival curves of the different mouse groups. A *p*-value < 0.05 was considered statistically significant.

## 3. Results

### 3.1. Construction of RHΔgra17Δnpt1 Strain

We previously demonstrated that live-attenuated *T. gondii* RH strain deficient in *gra17* (Figure 1A,C) was immunogenic in mice [13]. Interestingly, the NPT1 protein of *T. gondii* has been reported to play a key role in the growth and virulence of *T. gondii*, and the deletion of *npt1* gene in the RH strain also rendered the parasite immunogenic in mice [24]. We investigated whether a double mutant *T. gondii* strain deficient in *gra17* gene and *npt1* gene can be more immunogenic than mutant strain deficient in a single gene only. To accomplish this, a stable, single, and phleomycin-resistant strain of RHΔ*gra17*Δ*npt1* was successfully engineered by deleting the *npt1* gene in RHΔ*gra17* strain using CRISPR/Cas9-mediated non-homologous end joining (Figure 1B,D).

### 3.2. Characterization and Evaluation of RHΔgra17Δnpt1 Attenuation

We examined the impact of Δ*gra17*Δ*npt1* on the virulence of WT RH *T. gondii* in mice, and the results showed that RHΔ*gra17*Δ*npt1* completely reduced the pathogenicity of *T. gondii* RH strain. Mice infected with up to 10^6^ RHΔ*gra17*Δ*npt1* tachyzoites survived without developing clinical signs of toxoplasmosis. In contrast, mice infected with the WT tachyzoites became moribund and have died within eight days post-infection (Figure 2). These results show that double deletion of *gra17* and *npt1* genes significantly reduced the pathogenicity of the virulent *T. gondii* strain. To further examine the ability of these mutants to propagate *in vivo*, Kunming mice were infected with RHΔ*gra17*, RHΔ*npt1*, or RHΔ*gra17*Δ*npt1* strains (10^3^ tachyzoites/mouse, five mice/strain) and the parasite burden in the peritoneal fluid was evaluated by using the plaque assay five days after infection. Our data show that the ability of RHΔ*gra17*Δ*npt1* strain to grow and proliferate within HFF cells was significantly decreased compared to RHΔ*gra17* strain or RHΔ*npt1* strain (Figure 3), suggesting that the dual gene deletion has caused significant attenuation of the parasite virulence. Furthermore, at 60 days post-immunization, no parasite cysts were detected in the brain of mice immunized with RHΔ*gra17*Δ*npt1* strain.

### 3.3. Immune Responses Induced by Vaccination

In order to assess the immunogenicity of RHΔ*gra17*Δ*npt1*, the levels of specific anti-*T. gondii* IgG and IgG isotypes in the sera of vaccinated mice were determined using quantitative ELISA. At 60 days post-immunization, high levels of anti-*T. gondii* IgG, IgG1, and IgG2a were observed in the immunized mice compared with naïve, unvaccinated mice, suggesting that RHΔ*gra17*Δ*npt1* induced a strong humoral response (Figure 4). The cytokines produced by cultured mouse spleen cells stimulated by STAg were detected by ELISA. As shown in Figure 5, the levels of Th1-type cytokines (IFN-γ, IL-2, and IL-12) in the supernatants of STAg-stimulated splenocyte cultures of immunized mice were significantly higher than that of the naive mice. In addition, the level of Th2-type (IL-10) of immunized mice was significantly higher than that of naïve mice (Figure 5).

### 3.4. Protection Against Acute Infection

Mice vaccinated with 10^6^ RHΔ*gra17*Δ*npt1* were challenged with 10^3^ tachyzoites of type 1 strain RH or ToxoDB#9 strains (TgC7 or PYS) two months post-immunization. Mice were monitored for 30 days. High rates of survival were observed in all RHΔ*gra17*Δ*npt1*-immunized mice compared to unvaccinated mice. All mice that received vaccination survived to 30 days post-challenge with RH, PYS, or TgC7 strain. All unvaccinated + infected mice succumbed to the parasite challenge and died within 10 days after infection (Figure 6). These results indicate that vaccination using RHΔ*gra17*Δ*npt1* has conferred significant protection against lethal *T. gondii* infection with virulent homologous (RH) strain and heterologous (PYS and TgC7) strains.

### 3.5. Protection against Chronic Infection

Two months after immunization, mice were orally inoculated with cysts or oocysts of Pru strain in order to examine the protective efficacy of vaccination with RHΔ*gra17*Δ*npt1* against chronic infection. As expected, all unvaccinated mice infected with 100 cysts or oocysts succumbed to illness within 17 days (Figure 7A,B), whereas 75% and 62.5% of the unvaccinated mice infected with 10 cysts or oocysts had survived, respectively (Figure 7A,B). By contrast, all RHΔ*gra17*Δ*npt1*-vaccinated mice survived both 10 and 100 doses of cysts and oocysts (Figure 7). Parasite cyst burden in the survived mice was determined one month after infection. Unvaccinated mice challenged with a low dose of cysts or oocysts had 3215 ± 242 (cyst-infected group) and 8991 ± 1206 (oocyst-infected group) per brain, which was significantly higher than RHΔ*gra17*Δ*npt1*-vaccinated mice which had 275 ± 98 cysts/brain [cyst-infected group] and 461 ± 132 cysts/brain [oocyst-infected group]. In addition, mice vaccinated with RHΔ*gra17*Δ*npt1* challenged with a high dose (100 cysts or oocysts) had also significant reduction in the parasite cyst’s burden (cyst-infected group: 619 ± 211 cysts/brain; oocyst-infected group: 983 ± 225 cysts/brain), which was similar to RHΔ*gra17*Δ*npt1*-vaccinated mice challenged with the lower dose. These results show that RHΔ*gra17*Δ*npt1* can protect against chronic toxoplasmosis regardless of the parasite’s stage used to challenge the vaccinated mice.

### 3.6. Protection Against Congenital Infection

Pregnant mice were orally challenged with 10 *T. gondii* Pru oocysts on day 5 of gestation (each group contained six mice). Then, the abortion rate of pregnant mice, and the litter size and bodyweight of the neonates were recorded daily. Two dams from the unvaccinated + oocysts-infected group were euthanized at 14 and 17 days after infection due to inability to eat or drink, and their uterus showed severe pathological lesions (Figure 8A). Abortion was observed in the other four dams, including one dam that was euthanized at 21 days after infection due to inability to reach food or water. In contrast, no abortions were detected in the dams from RHΔ*gra17*Δ*npt1* vaccinated + oocyst-infected group and unvaccinated + uninfected mice, and all their pups were successfully delivered. However, the litter size and bodyweight of neonates of unvaccinated + uninfected dams were significantly larger compared to those of RHΔ*gra17*Δ*npt1* vaccinated + oocyst-infected dams (Figure 8B,C).

One month after birth, pups and dams from RHΔ*gra17*Δ*npt1*-vaccinated + oocyst-infected mice and the rest of the dams from unvaccinated + oocyst-infected mice were euthanized, and their brain cyst’s burden was determined. The results showed that brain cysts (227 ± 138) were detected in 77.8% (42/54) of the pups. Also, RHΔ*gra17*Δ*npt1*-vaccinated + oocyst-infected dams (703 ± 179 cysts/brain; six mice) had significantly less brain cyst’s burden than unvaccinated + oocyst-infected dams (11139 ± 1197 cysts/brain; 3 mice).

## 4. Discussion

Previous studies have shown that *gra17* is essential for maintaining the structural integrity of the PV, and mediating the trafficking of substances across the PVM, whereas *npt1* is essential for the survival and virulence of *T. gondii* [16,17]. Based on these facts, we hypothesized that attenuated *T. gondii* RH strain harboring double deletions in *gra17* gene and *npt1* gene could elicit a safer and more protective immune response against *T. gondii* infection than the strain with a single deletion only. In this study, the RHΔ*gra17*Δ*npt1* strain was successfully engineered using CRISPR/Cas9 method. Our results showed that the deletion of the *gra17* gene and the *npt1* gene resulted in a significant attenuation of the virulence of RH strain, as revealed by the significant difference in the survival rates and signs of illness of mice infected with mutant strain *versus* the WT strain. Vaccination using an infection dose up to 10^6^ RHΔ*gra17*Δ*npt1* achieved a balance between safety and immunogenicity and was, therefore, used in all vaccination experiments.

Consistent with other live-attenuated *T. gondii* vaccines, mice vaccinated with RHΔ*gra17*Δ*npt1* exhibited a high level of anti-*T. gondii* IgG antibodies [13,14,24,25]. These specific IgG antibodies play an important role in protection against *T. gondii* infection via hindering the attachment of the parasite to the host cells, limiting its ability to establish infection, and activation of the classical complement pathway [26]. Our results showed that RHΔ*gra17*Δ*npt1*-vaccinated mice developed a strong humoral immune response, as shown by the high level of IgG1 and IgG2a antibodies compared to unvaccinated mice at two months post-immunization. The patterns of cytokines in the supernatant of cultured spleen cells showed that Th1 cytokines (IFN-γ, IL-2, and IL-12) and Th2 cytokines (IL-10) in RHΔ*gra17*Δ*npt1*-vaccinated mice were significantly higher than those in unvaccinated mice.

*T. gondii* infection can produce IFN-γ and IL-12-dependent cell-mediated immune response in order to inhibit tachyzoite’s proliferation [27,28]. The correlation between high levels of Th1-type cytokines and strong humoral immune response, and vaccination, in addition to the known role of Th1 cytokines in *T. gondii* infection strongly suggest that Th1-type cytokines elicited by RHΔ*gra17*Δ*npt1* might contribute to the protection of mice against a lethal challenge with WT strains (type 1 RH or two ToxoDB#9 strains). Also, we detected a high level of Th2 cytokine IL-10, which is essential for balancing the excessive inflammatory response associated with acute *T. gondii* infection [29]. Next, we explored whether the balance of Th1 and Th2 cytokines in RHΔ*gra17*Δ*npt1*-vaccinated mice was effective in protection against chronic *T. gondii* infection. As expected, the immune response induced by RHΔ*gra17*Δ*npt1* vaccination significantly protected mice against different doses (10 and 100) of type II Pru cysts and oocysts. Our data clearly demonstrate the ability of RHΔ*gra17*Δ*npt1* to significantly reduce 91.4% of the parasite cyst burden compared to unvaccinated mice. In previous studies, it was demonstrated that vaccination using single-gene mutants, RHΔ*gra17* [13] and RHΔ*npt1* [24] resulted in 98.8% and 98.6% reduction in the parasite cyst burden, respectively. This finding suggests that although deletion of both *gra17* and *npt1* genes has considerably attenuated the virulence of the parasite, this was not paralleled by improvement in the double mutant’s ability to limit the parasite cyst burden during latent infection.

We also examined the protective efficacy of RHΔ*gra17*Δ*npt1* against congenital toxoplasmosis. We simulated *T. gondii* infection in the first trimester, where pregnant mice were orally infected with 10 oocysts at day 5 of gestation. Unvaccinated + oocyst-infected pregnant mice had abortion, and three dams could not eat or drink, and showed severe clinical signs of toxoplasmosis, and therefore were euthanized. This result agrees with a previous report of abortion in infected mice during the first trimester [30]. By contrast, mice from RHΔ*gra17*Δ*npt1*-vaccinated + oocyst-infected and unvaccinated + uninfected groups delivered their pups successfully. Moreover, a significantly lower cyst burden in the brain was observed in RHΔ*gra17*Δ*npt1*-vaccinated dams compared to unvaccinated + oocyst-infected dams. However, the litter size and bodyweight of neonates from RHΔ*gra17*Δ*npt1*-vaccinated + oocyst-infected mice were significantly lower than that of unvaccinated + uninfected mice.

The high levels of Th1-type cytokine responses (IFN-γ, IL-2, and IL-12) protect pregnant mice against *T. gondii* infection and limit congenital transmission. However, during pregnancy, the maternal immune response tends to be of the Th2 humoral type in order to sustain the progress of pregnancy [31]. While beneficial for the pregnancy, Th2-dominated immune response can increase the risk of *T. gondii* infection in pregnant mice. The increase of Th2-related IL-10, an antiinflammatory cytokine produced by many subsets of activated immune cells, is interesting because it plays a key role in down-regulating IFN-γ responses in *T. gondii*-infected mice [13]. Previous studies indicated that IL-10-deficient C57BL/6 mice infected with *T. gondii* exhibited a high level of IL-12, which increases the levels of IFN-γ and TNF-α, and eventually led to hepatic inflammation and necrosis [29]. In addition, the pregnancy outcome can be improved in C57BL/6 mice infected with *T. gondii* by treatment with recombinant IL-10 and can be deteriorated in IL-10-deficient mice compared to naïve mice [32]. Therefore, to ensure successful pregnancy and reduce maternal susceptibility to *T. gondii*, a balanced Th1/Th2 immune response is essential. In the present study, we showed that RHΔ*gra17*Δ*npt1* vaccination induced a mixed Th1/Th2 immune response in mice and improved the pregnancy outcome compared to unvaccinated + infected mice. However, RHΔ*gra17*Δ*npt1* immunization did not completely block the maternal–fetal transmission, as indicated by the lower litter size and mean body weight of pups born to vaccinated + infected dams compared to pups born to unvaccinated + uninfected mice. It is possible that RHΔ*gra17*Δ*npt1* is overattenuated and cannot persist longer in the host; consequently, was unable to fully protect the pups. More understanding of how the proinflammatory and antiinflammatory cytokines are regulated post-immunization will advance our understanding of how RHΔ*gra17*Δ*npt1* activates host immune responses in the vaccinated mice.

## 5. Conclusions

The data reported in this study show that attenuated double-deletion mutant derived from the virulent *T. gondii* type 1 RH strain is safe and immunogenic, and provides protection in mice against virulent *T. gondii* challenge. A single i.p. vaccination with 10^6^ RHΔ*gra17*Δ*npt1* tachyzoites provided protection against acute infection with the homologous and heterologous *T. gondii* strains. Also, RHΔ*gra17*Δ*npt1* mutant strain generated a protective immune response that increased the survival rate and reduced parasite cyst burden in the brain of chronically infected mice. Mice vaccinated with RHΔ*gra17*Δ*npt1* showed partial protection against congenital transmission of *T. gondii*, where significantly reduced cyst burdens were detected in the brain of the vaccinated dams and their pups. Vaccination also improved the litter size and body weight of pups born to vaccinated dams compared to those born to unvaccinated infected dams. These data show that RHΔ*gra17*Δ*npt1* can be a promising live-attenuated vaccine candidate against toxoplasmosis. Further research to understand the immunological mechanisms responsible for the protection and to uncover the correlates of protection is warranted and will be critical for advancing the development of toxoplasmosis vaccines.

## Figures and Tables

**Figure 1 microorganisms-08-00352-f001:**
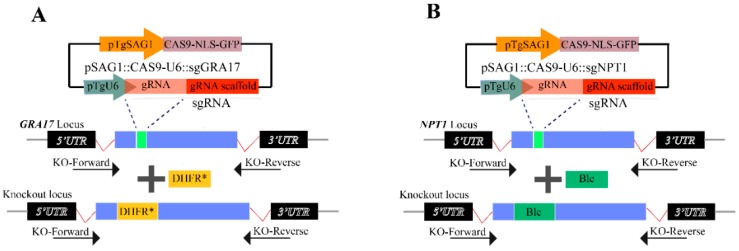
Generation of double *gra17* and *npt1* knock-out strains of *T. gondii* Type 1 RH strain using CRISPR/Cas9. (**A**) A schematic illustration of CRISPR/Cas9 system used in disrupting the *gra17* gene by insertion of pyrimethamine-resistant DHFR (DHFR*) cassette. (**B**) Schematic showing deletion of *npt1* gene by insertion of a phleomycin-resistant Ble into *npt1* gene. The KO-forward and KO-reverse primers were used to amplify the small fragment. (**C**) Diagnostic PCR showing that a small fragment was lost due to insertion of the DHFR* cassette with short extension time in the *gra17* mutant strain compared to the wild-type strain. (**D**) Diagnostic PCR showing the successful knockout of the *npt1* gene in the RHΔ*gra17* strain.

**Figure 2 microorganisms-08-00352-f002:**
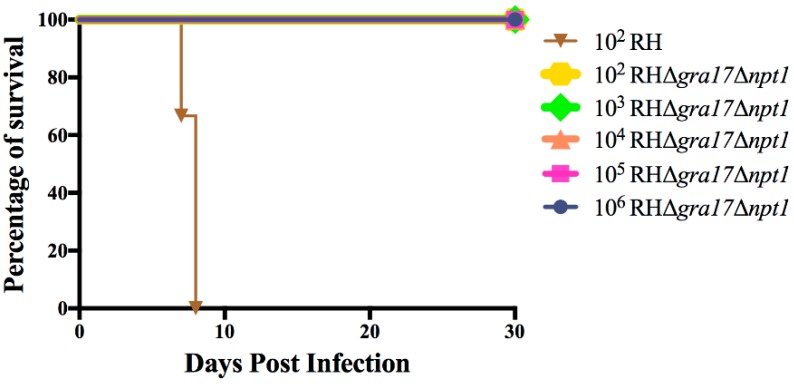
Virulence assays of RHΔ*gra17*Δ*npt1* in Kunming mice. Serial doses (10^2^ to 10^6^) of RHΔ*gra17*Δ*npt1* or 10^2^ of wild type RH tachyzoites were i.p. inoculated into mice, and the survival rates and clinical signs were recorded for 30 days. Six mice were used per group.

**Figure 3 microorganisms-08-00352-f003:**
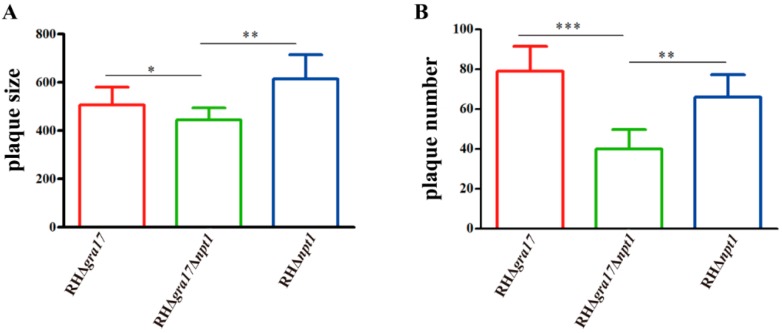
In vivo and in vitro growth of RHΔ*gra17*Δ*npt1*. Approximately 10^3^ tachyzoites of RHΔ*gra17*, RHΔ*npt1*, or RHΔ*gra17*Δ*npt1* were i.p. inoculated into mice. Five days later, the parasite burden in the peritoneal fluid was evaluated using the in vitro plaque assay. The data represent the mean ± standard deviation (SD) of the relative size (**A**) and number (**B**) of the plaques generated by tachyzoites of each of the three tested mutant strains. Considerable attenuation of RHΔ*gra17*Δ*npt1* was demonstrated by the significant reduction in the size and number of plaques produced by this double mutant strain RHΔ*gra17*Δ*npt1* compared to that produced by RHΔ*gra17* strain or RHΔ*npt1* strain. Statistically significant differences are indicated by *, *p* < 0.05; **, *p* < 0.01; ***, *p* < 0.001.

**Figure 4 microorganisms-08-00352-f004:**
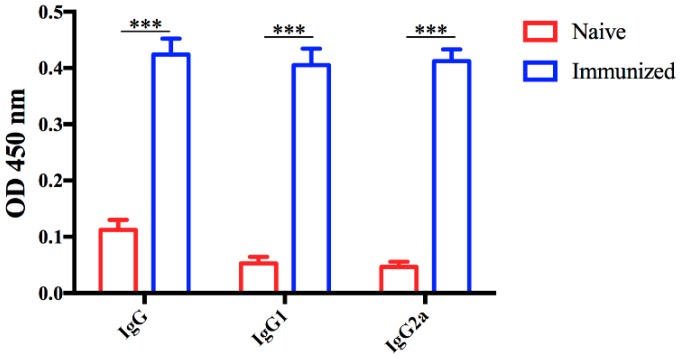
Serum antibody levels in mice vaccinated with RHΔ*gra17*Δ*npt1*. The levels of total IgG and IgG isotypes (IgG1 and IgG2a) were determined in the serum of Kunming mice vaccinated with RHΔ*gra17*Δ*npt1*. Analysis was performed by using ELISA at 60 days post-vaccination. Results are expressed as mean of OD_450_ ± SD, and level of significance was compared to control naïve mice (***, *p* < 0.001).

**Figure 5 microorganisms-08-00352-f005:**
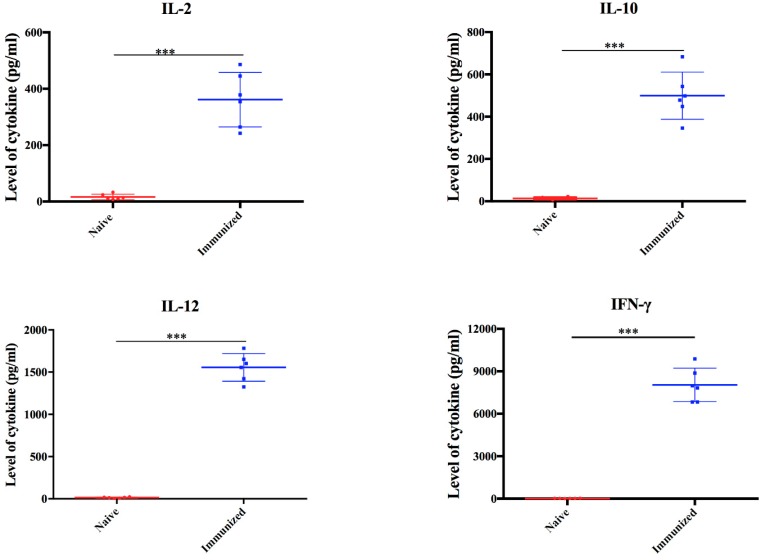
The levels of cytokines detected in the supernatant of splenocyte cultures of RHΔ*gra17*Δ*npt1*-vaccinated mice. Spleen cells were harvested from vaccinated and naïve mice at 60 days post-immunization and stimulated in vitro with 10 µg/mL STAg. Evaluation of Th1 (IL-2, IL-12 and IFN-γ) and Th2 (IL-10) cytokines in the splenocyte culture supernatants was performed using ELISA. Data points are represented as means ± SD. A statistically significant difference is indicated by ***, *p* < 0.001.

**Figure 6 microorganisms-08-00352-f006:**
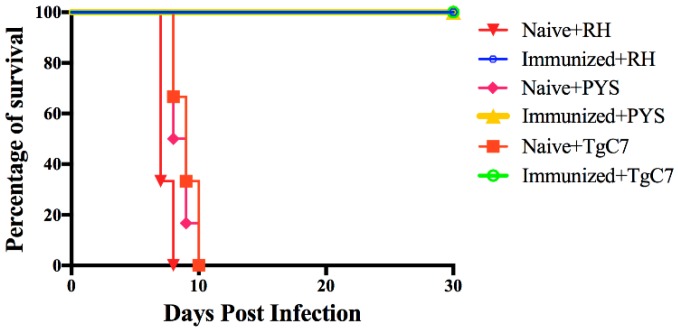
Survival curves of vaccinated and infected mice. At 60 days post-vaccination, mice were i.p. challenged with 10^3^
*T. gondii* tachyzoites of RH, PYS, or TgC7 strains (6 mice/strain), and monitored for 30 days. Survival curves of RHΔ*gra17*Δ*npt1*-vaccinated mice showed a high survival rate compared to naïve mice.

**Figure 7 microorganisms-08-00352-f007:**
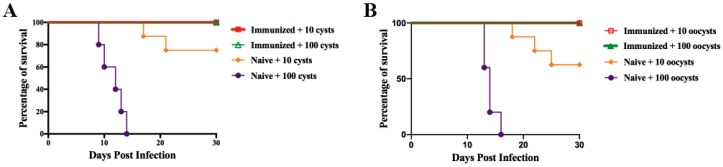
Protective effect of RHΔ*gra17*Δ*npt1* vaccination against chronic *T. gondii* infection using tissue cysts or oocysts. Survival rates following challenge of RHΔ*gra17*Δ*npt1*-vaccinated mice with (**A**) 10 or 100 tissue cysts, or (**B**) 10 or 100 oocysts of type 2 Pru strain at 60 days post-vaccination over 30 days. All RHΔ*gra17*Δ*npt1*-vaccinated mice survived regardless of the inoculation dose or the parasite stage (cyst or oocyst) compared to naive mice. Eight mice per group were infected with 10 cysts or oocysts, and five mice per group were infected with 100 cysts or oocysts.

**Figure 8 microorganisms-08-00352-f008:**
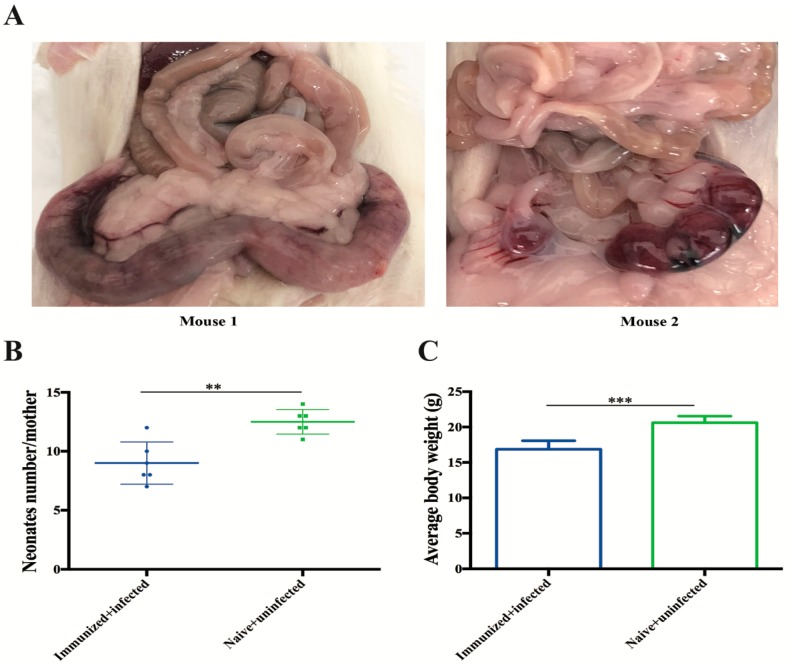
Protection of RHΔ*gra17*Δ*npt1*-vaccinated mice against type II Pru oocyst infection on day 5 of gestation. Two mice from the unvaccinated + oocyst-infected group were unable to eat or drink and were euthanized on days 14 (Mouse 2) and 17 (Mouse 1) after infection. Significant pathologic lesions were observed in their uterus (**A**). The litter size (**B**) and average body weight (**C**) of pups born to unvaccinated + uninfected mice *versus* RHΔ*gra17Δnpt1*-vaccinated mice + infected mice were assessed at 30 days after birth. Data points are represented as means ± SD. Statistically significant difference compared with unvaccinated + uninfected (naïve) mice: **, *p* < 0.01, ***, *p* < 0.001.

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
