# Peer review of "RHΔgra17Δnpt1 Strain of Toxoplasma gondii Elicits Protective Immunity Against Acute, Chronic and Congenital Toxoplasmosis in Mice"

_microorganisms, 2020, doi:10.3390/microorganisms8030352_

Round 1

Reviewer 1 Report

Report on Microorganisms-715795

This paper describes an attenuated version, created by Crispr-Cas, of the lethal T. gondii strain, RH, deficient in two metabolic genes. This strain is tested as an immunogen to prime for resistance against the virulent wild type RH and two other virulent strains. The strain proves to function as a vaccine, inducing high levels of antibody and inflammatory cytokines. The strain is also used in a model to mimic first-trimester congenital infection. The model is successful, rescuing pregnant females from major pathology and allowing live birth. However the progeny are small and light and mostly carry brain cysts.

Corrections:

In general, the paper is written in good English. I have indicated a few corrections. The following data are missing: numbers of mice used in all the in vivo In one case the number is stated as 8 animals per group, but 1 and possibly two groups almost certainly contained only 5 animals. Accurate numbers must be given for all groups. In survival curves, several groups were 100% surviving: in these cases the colours of 3 different lines are overlaid and therefore invisible. The lines should be separated so that the different colours can be seen.

Missing experiments

The data lack experimental evidence of the ability of the attenuated strain to replicate either in vitro or in vivo. These experiments must be supplied, the experiments quantitatively analysed, and the data presented. The authors do not ask whether the attenuated strain is itself cystogenic. This must be investigated. RH is often called “non-cystogenic” but this is not always assayed in situations where it becomes avirulent, as this. No data is presented for the results of virulence assays referred to in 3.1 in the manuscript. The data must be shown.

Author Response

Responses to comments and suggestions of Reviewer #1:

General comments:

This paper describes an attenuated version, created by Crispr-Cas9, of the lethal T. gondii strain, RH, deficient in two metabolic genes. This strain is tested as an immunogen to prime for resistance against the virulent wild type RH and two other virulent strains. The strain proves to function as a vaccine, inducing high levels of antibody and inflammatory cytokines. The strain is also used in a model to mimic first-trimester congenital infection. The model is successful, rescuing pregnant females from major pathology and allowing live birth. However, the progeny are small and light and mostly carry brain cysts.

Response: We thank reviewer #1 very much for favorable comments and constructive suggestions on our MS. We have also revised the MS strictly according to the reviewer’s comments and suggestions as we detail in the following points.

Main comments:

Point 1: In general, the paper is written in good English. I have indicated a few corrections. The following data are missing: numbers of mice used in all the in vivo. In one case the number is stated as 8 animals per group, but 1 and possibly two groups almost certainly contained only 5 animals. Accurate numbers must be given for all groups. In survival curves, several groups were 100% surviving: in these cases the colours of 3 different lines are overlaid and therefore invisible. The lines should be separated so that the different colures can be seen.

Responses: We thank reviewer #1 very much for constructive suggestions on our MS. We have added the accurate numbers of mice in each group and highlighted the overlaid lines of survival curves into the revised MS in order to make it clearer to the readers.

Point 2: The data lack experimental evidence of the ability of the attenuated strain to replicate either in vitro or in vivo. These experiments must be supplied, the experiments quantitatively analysed, and the data presented. The authors do not ask whether the attenuated strain is itself cystogenic. This must be investigated. RH is often called “non-cystogenic” but this is not always assayed in situations where it becomes avirulent, as this. No data is presented for the results of virulence assays referred to in 3.1 in the manuscript. The data must be shown.

Responses: We thank reviewer #1 very much for constructive suggestions on our MS. We have added the suggested data of parasites growth assays in vitro in the revised MS. Also, two months after immunization with RHΔgra17Δnpt1 tachyzoites, the mice were humanely sacrificed in order to detect the cysts in the mouse brain by microscope examination. Our resulted showed that there were no cysts in the brain of the immunized mice.

Reviewer 2 Report

This manuscript proposes to evaluate the protective efficacy of vaccination using RHΔgra17Δnpt1 tachyzoites against acute, chronic and congenital Toxoplasma gondii toxoplasmosis. It is a highly relevant study since the development of an effective and safe vaccine is an important aim due to the great clinical and economic impact of this parasitosis. The experiments were well designed, and the results presented are very solid.

The current manuscript is a following study that uses a developed double deletion mutant. The same group previously published two manuscripts using the single mutants RHΔgra17 and RH:1NPT1 strains. The results obtained in the current manuscript show that RHΔgra17Δnpt1 can provide robust protection against acute (100% of protection) and chronic infection of toxoplasmosis regardless of the parasite’s stage used to challenge the immunized mice. However, the use of the double mutant does not seem to provide additional protection compared to the single mutant strains which already showed 100% of protection.  Also, this new attenuated parasite didn´t improve the protective effect against cyst formation compared to previous published RH:1NPT1 mutant strain

Questions that should be clarified:

1) IL-10 is typically a regulatory cytokine secreted by many different cell types. In addition to Th2 cells, IL-10 is known to be produced by many cell types including Tr1 cells, CD25+ and CD25- T regulatory cells, Th1 cells, macrophages and dendritic cells as well as B lymphocytes. If authors want to evaluate a Th2 immune response they must study IL-4 or IL-5 secretion

2) It is not clear from where the authors got the oocysts. In Materials and Methods section it is only explained that cysts of T. gondii type 2 Pru strain were maintained in Kunming mice. Nothing is mentioned about oocysts. Oocysts are only isolated from cat faeces.

3) Statistic is missing in the comparison of cyst burden (section 3.4, page 12, line 245)

4) In the statement “Also, RHΔgra17Δnpt1-vaccinated + oocyst-infected dams (11139 ± 1197 cysts/brain) had significantly less brain cyst burdens than non-vaccinated + oocyst infected dams (703 ± 179 cysts/ brain).” I suppose that they made a mistake and 11139 ± 1197 cysts/brain refers to non-vaccinated + oocyst infected dams and 703 ± 179 cysts/ brain to vaccinated dams

5) In Discussion Section: Line 324, authors state that “the parasite cyst burden was still higher than that in mice immunized with 325 single RHΔgra17 gene or RHΔnpt1 deletion [13, 22]” To compare with previous results, the percentage of reduction against the unimmunized should be used, not the absolute number of cysts.

6) Discussion Section: I think that finding high levels of Th1 cytokines along with a strong protection is not enough to draw the conclusion that “High levels of Th1-type cytokines produced by vaccination with RHΔgra17Δnpt1 effectively protected mice against a lethal challenge with WT strains (type 1 RH or two ToxoDB#9 strains)”. I would say that the correlation between both parameters in addition to the known role of Th1 cytokines in T. gondii infection strongly suggest that Th1-type cytokines elicited by the immunization might contribute to the protection obtained

Author Response

Responses to comments and suggestions of Reviewer #2:

General comments:

This manuscript proposes to evaluate the protective efficacy of vaccination using RHΔgra17Δnpt1 tachyzoites against acute, chronic and congenital Toxoplasma gondii toxoplasmosis. It is a highly relevant study since the development of an effective and safe vaccine is an important aim due to the great clinical and economic impact of this parasitosis. The experiments were well designed, and the results presented are very solid. The current manuscript is a following study that uses a developed double deletion mutant. The same group previously published two manuscripts using the single mutants RHΔgra17 and RH:1NPT1 strains. The results obtained in the current manuscript show that RHΔgra17Δnpt1 can provide robust protection against acute (100% of protection) and chronic infection of toxoplasmosis regardless of the parasite’s stage used to challenge the immunized mice. However, the use of the double mutant does not seem to provide additional protection compared to the single mutant strains which already showed 100% of protection.  Also, this new attenuated parasite didn´t improve the protective effect against cyst formation compared to previous published RHΔNPT1 mutant strain.

Responses: We thank reviewer #2 very much for favorable comments and constructive suggestions on our MS. We appreciate the important observation made by the reviewer about the relevance of using single or double mutant strains, well noted.

Main comments:

Point 1: IL-10 is typically a regulatory cytokine secreted by many different cell types. In addition to Th2 cells, IL-10 is known to be produced by many cell types including Tr1 cells, CD25+ and CD25- T regulatory cells, Th1 cells, macrophages and dendritic cells as well as B lymphocytes. If authors want to evaluate a Th2 immune response they must study IL-4 or IL-5 secretion

Responses: We thank reviewer #2 very much for constructive suggestions on our MS. Indeed, the levels of IL-4 and IL-5 must be used to evaluate a Th2 immune response. Unfortunately, in this study we only evaluated the level of IL-10. However, our previous studies and other previous studies of live attenuated strain vaccines showed that IL-4 and IL-5 were not significantly increased. In the revised version of the present MS, we have revised the discussion to reflect the need to measure more cytokines, not just IL-10 when assessing immune response post vaccination. We stated that more understanding of how the proinflammatory and anti-inflammatory cytokines are regulated post immunization will advance our understanding of how RHΔgra17Δnpt1 activates host immune responses in the vaccinated mice. We also indicated that an anti-inflammatory cytokine IL-10 can be produced by many subsets of activated immune cells.

Point 2: It is not clear from where the authors got the oocysts. In Materials and Methods section it is only explained that cysts of T. gondii type 2 Pru strain were maintained in Kunming mice. Nothing is mentioned about oocysts. Oocysts are only isolated from cat faeces.

Responses: We thank reviewer #2 very much for constructive suggestions on our MS. We have added the missed information about oocyst production and purification in the Materials and Methods section.

Point 3: Statistic is missing in the comparison of cyst burden (section 3.4, page 12, line 245)

Responses: We thank reviewer #2 very much for constructive comments on our MS. We have added this part of the statistics as suggested.

Point 4:  In the statement “Also, RHΔgra17Δnpt1-vaccinated + oocyst-infected dams (11139 ± 1197 cysts/brain) had significantly less brain cyst burdens than non-vaccinated + oocyst infected dams (703 ± 179 cysts/ brain).” I suppose that they made a mistake and 11139 ± 1197 cysts/brain refers to non-vaccinated + oocyst infected dams and 703 ± 179 cysts/ brain to vaccinated dams.

Responses: We thank reviewer #2 very much for constructive comments on our MS. You are correct, we are very sorry for this oversight. This has been corrected accordingly.

Point 5: In Discussion Section: Line 324, authors state that “the parasite cyst burden was still higher than that in mice immunized with single RHΔgra17 gene or RHΔnpt1 deletion [13, 22]” To compare with previous results, the percentage of reduction against the unimmunized should be used, not the absolute number of cysts.

Responses: We thank reviewer #2 very much for constructive comments on our MS. We have added the reduction rate of the cyst burden of RHΔgra17Δnpt1, RHΔgra17 and RHΔnpt1 into the revised MS as recommended.

Point 6: Discussion Section: I think that finding high levels of Th1 cytokines along with a strong protection is not enough to draw the conclusion that “High levels of Th1-type cytokines produced by vaccination with RHΔgra17Δnpt1 effectively protected mice against a lethal challenge with WT strains (type 1 RH or two ToxoDB#9 strains)”. I would say that the correlation between both parameters in addition to the known role of Th1 cytokines in T. gondii infection strongly suggest that Th1-type cytokines elicited by the immunization might contribute to the protection obtained.

Responses: We thank reviewer #2 very much for constructive comments on our MS. We have revised this sentence accordingly.

We have done our best to address all comments and we sincerely hope that you find our MS revised to your satisfaction. We are looking forward to receiving your editorial decision soon and hope to see our work published in Microorganisms.

With best wishes
